

# A review of deep learning in blink detection

Jianbin Xiong[1,*], Weikun Dai[1,*], Qi Wang[1], Xiangjun Dong[1],
Baoyu Ye[2] and Jianxiang Yang[1]

[1] School of Automation, Guangdong Polytechnic Normal University, Guangzhou, Guangdong, China
[2] School of Aircraft Maintenance Engineering, Guangzhou Civil Aviation College, Guangzhou, Guangdong, China
* These authors contributed equally to this work.

## ABSTRACT

Blink detection is a highly concerned research direction in the field of computer vision, which plays a key role in various application scenes such as human-computer interaction, fatigue detection and emotion perception. In recent years, with the rapid development of deep learning, the application of deep learning techniques for precise blink detection has emerged as a significant area of interest among researchers. Compared with traditional methods, the blink detection method based on deep learning offers superior feature learning ability and higher detection accuracy. However, the current research on blink detection based on deep learning lacks systematic summarization and comparison. Therefore, the aim of this article is to comprehensively review the research progress in deep learning-based blink detection methods and help researchers to have a clear understanding of the various approaches in this field. This article analyzes the progress made by several classical deep learning models in practical applications of eye blink detection while highlighting their respective strengths and weaknesses. Furthermore, it provides a comprehensive summary of commonly used datasets and evaluation metrics for blink detection. Finally, it discusses the challenges and future directions of deep learning for blink detection applications. Our analysis reveals that deep learning-based blink detection methods demonstrate strong performance in detection. However, they encounter several challenges, including training data imbalance, complex environment interference, real-time processing issues and application device limitations. By overcoming the challenges identified in this study, the application prospects of deep learning-based blink detection algorithms will be significantly enhanced.

## INTRODUCTION

With the rapid development and application of artificial intelligence and human-computer interaction technology (*Ren & Bao, 2020*), human facial information has received a lot of attention. As one of the most important parts of human face, the study on the characteristics of eyes is a hot spot and a great challenge (*Bisogni et al., 2024*). The act of blinking is a crucial characteristic of the eye, serving as an effective indicator of an

Corresponding authors
Qi Wang, aotomwq@sina.com
Xiangjun Dong, dxj0801@163.com

individual's current physiological and psychological state (*Dewi et al., 2022*; *Hong et al., 2024*). Moreover, it holds significant importance as a control signal in applications related to human-computer interaction (*Hu et al., 2023*).

Due to the rapid development of computer vision technology in recent years, it has led to the advancement of blink detection technology (*Phuong et al., 2022*). Blink detection begins to be applied to a variety of fields in production and life practice, such as medicine, transportation, security and so on. In the medical field, blink detection can be applied to dry eye recovery (*Muntz et al., 2022*). One of the main causes of dry eye syndrome is a reduction in the number of blinks, and monitoring a target's blink rate can help relieve dry eye by reminding the target to blink. In the traffic field, blink tracking is an important component of driver assistance systems. As an important landing place of artificial intelligence technology, assisted driving has huge economic benefits. Blink detection monitors the driver's fatigue by detecting the driver's current blink frequency (*Ibrahim et al., 2021*; *Seha et al., 2021*). This ensures that the driver is mentally alert when driving and reduces the likelihood of accidents. In the security field, blink detection can be used to improve the reliability of face recognition authentication software (*Masood et al., 2023*). Criminals can fake facial videos through artificial intelligence (AI) technology. Usually, the people in these fake videos do not blink, so whether they blink can be used as the basis for judging the authenticity of the video. In addition, blink detection can also be applied to help disabled people interact with computers (*Xiong et al., 2013*). Under normal circumstances, users can control computer systems through conventional devices such as mouse, keyboard, touch screen, *etc*. However, for some people with disabilities, such as patients with symptoms of paralysis and muscular dystrophy, relying on contact devices cannot provide for their normal human-computer interaction needs. Therefore, the human-computer interaction technology based on blink detection has been the focus of researchers (*Attiah & Khairullah, 2021*; *Medeiros et al., 2022*; *Neogi, Das & Deb, 2021*). This technology can give people with disabilities the opportunity to enjoy the convenience brought by artificial intelligence, so that they can better integrate into the information-based society. Therefore, the accurate detection of blinking as a characteristic information or control signal holds significant practical application value and promising market prospects. At the same time, with the increasing demand of natural and comfortable non-contact blink detection system, the research on blink detection can produce good economic and social benefits.

In Table 1, the studies on blink detection by collecting information from the eye in different modalities are summarized. Through the survey, the mainstream methods for blink detection technology can be categorized into two main types: contact blink detection and non-contact blink detection. Figure 1 has summarized the currently common methods of blink detection. Contact blink detection depends on the subject wearing a wearable sensor to detect blinking movements. In contrast, non-contact blink detection utilizes external sensory signals without requiring the subject to wear any contact sensors. This method detects dynamic information about the target by analyzing changes in the features of these sensory signals. Contact blink detection mainly uses bioelectrical signals (such as electrooculography (EOG) (*Reddy et al., 2011*; *Tag et al., 2016*) and infrared reflective

**Table 1  Literature related to different modalities of blink detection.**

| Modalities | Definition | Number of literatures |
|---|---|---|
| Images | Methods such as facial landmark detection, eye aspect ratio (EAR) computation, or deep learning models are used to analyze the eye region in an image to determine the opening and closing states of the eyes and to recognize blinking actions. | 12 |
| Videos | Eye regions in consecutive video frames are tracked and analyzed to recognize blinking movements. | 13 |
| EOG | Blinking was detected by measuring electrical potential changes generated by eye movements through electrode pads attached to the eye periphery of the subject. | 6 |
| Infrared reflective | Using infrared light to illuminate the eye and a light-sensitive diode to receive the reflected light to obtain a blinking signal. | 4 |

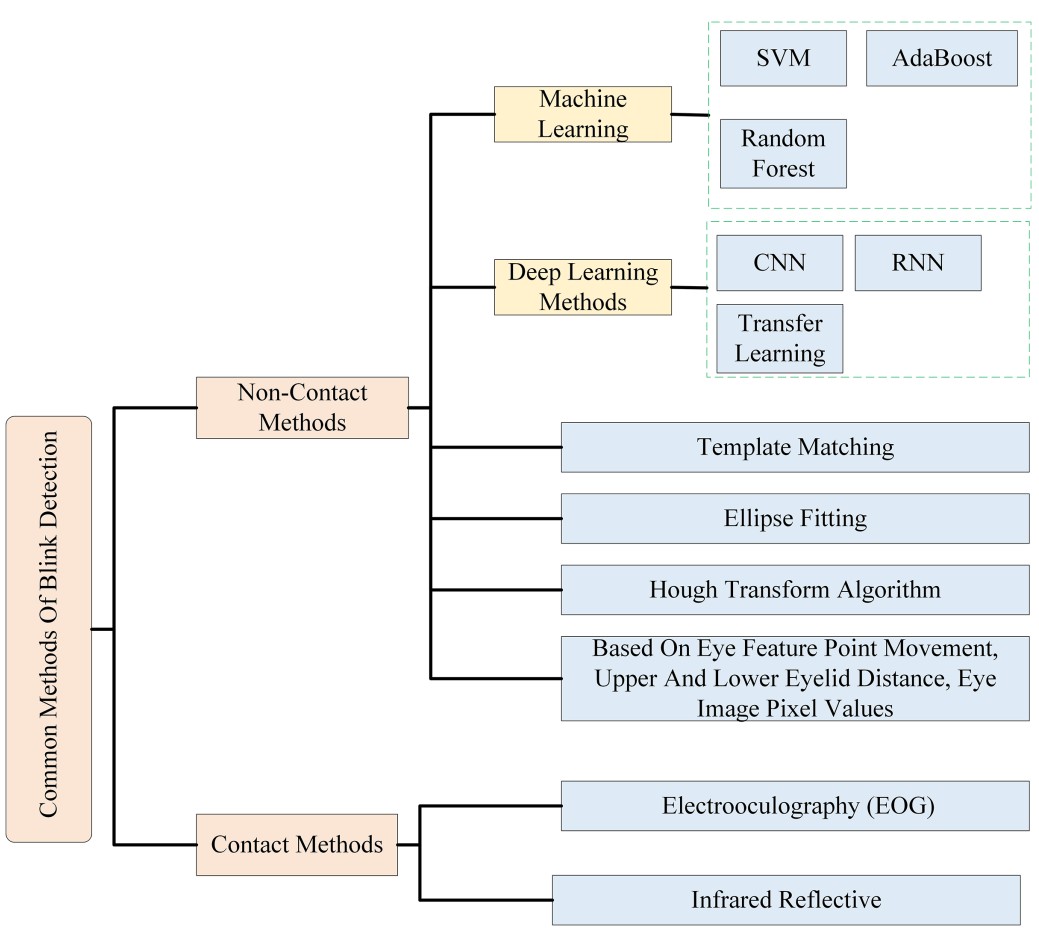

**Figure 1  Classification of commonly used methods for blink detection.**

sensors (*Dementyev & Holz, 2017*) to detect blinks. EOG are obtained by measuring changes in electrical potentials generated by eye movements with electrode pads attached around the subject's eyes (*Bulling et al., 2011*). As early as 1998, *Kong & Wilson (1998)* proposed an efficient and robust blink detection algorithm based on nonlinear analysis of EOG signals. However, it suffers from the problem that its application scenarios are limited

to those in the laboratory. In 2016, *Rodriguez et al. (2016)* proposed a new method based on bioelectrical signals. The detection is realized by extracting the electrical signal of the eye muscle, which is a new method for contact blink detection. Infrared sensors, a very popular technology, have been used in the field of blink detection (*Dementyev & Holz, 2017*). Infrared blink detection is usually achieved by installing an infrared reflective diode coupled with an infrared photodiode on an eyewear, which uses modulation of the infrared beam to efficiently capture head movements and changes in environmental light, resulting in fast and accurate blink detection (*Ishimaru et al., 2014*). The sensors required for contact blink detection are in direct contact with the person and severely limit normal activity, and it is unrealistic to achieve real-time detection in life in this way. Therefore, non-contact blink detection methods are now the mainstream.

Most non-contact blink detection efforts are realized by computer vision methods (*Królak & Strumiłło, 2012*; *Saealal et al., 2022*; *Xiong et al., 2016*). In the early years, non-contact blink detection was mainly determined by the appearance characteristics of the eyes (*Marcos-Ramiro et al., 2014*) or the motion characteristics of the eye pixels (*Drutarovsky & Fogelton, 2015*). These methods are sensitive to changes in scale and rotation of the eye image. When the image is scaled or rotated, the accuracy of the detection decreases. Moreover, it is necessary to make rules and determine some specific thresholds manually, and the generalization is difficult to be guaranteed. With the application of machine learning, the above problems are effectively improved. In 2010, *Lee, Lee & Park (2010)* detected blinking by using support vector machine (SVM) as a blink classifier and using AdaBoost to detect the eye region of the input image. In 2014, *Marcos-Ramiro et al. (2014)* proposed a randomized forest-based blink detection method. A face tracker is first used to localize the eyes. Then, a random forest classifier is applied to the eye region to associate each pixel with each of the three eye parts: pupil, sclera, and eyelid. For each frame, the eye opening (the number of pixels classified as pupil and sclera) is calculated, as with the difference in eye opening compared to the previous two frames. The values are compared to a threshold value to identify whether or not a blink has occurred. This method is able to achieve the F1-score of 93.65%, but the random forest algorithm is relatively computationally complex and consumes a long time because it requires the construction of multiple decision trees and the aggregation of results. Machine learning methods have improved in robustness and generalization ability compared to earlier methods. However, when using machine learning methods as classifiers, they need to rely on a manually designed feature extraction process, which requires a great deal of expertise and experience to identify and construct effective features. Therefore, it is difficult to find an optimal feature extraction method to extract image features as input to the classifier (*Kayadibi et al., 2022*; *Zhao et al., 2018*).

With the continuous development of technology, network structures with a shallow number of layers gradually cannot meet the need of research (*Xiong, Nie & Li, 2022*). Researchers began to study the deep network structure, and a large number of deep learning models appeared (*Dong, Wang & Abbas, 2021*; *Shlezinger et al., 2023*), such as: convolutional neural networks (CNN), recurrent neural networks (RNN), deep belief networks (DBN), *etc*. Compared with traditional detection algorithms, deep learning has

obvious advantages in blink detection. Firstly, deep learning models have deeper networks with strong feature extraction capabilities. By training the neural network, the model can gradually acquire various features from low to high level, such as edges, textures, shapes and so on. Secondly, deep learning methods can not only handle massive data, but also continuously learn new knowledge as the amount of data increases. Blink detection involves many different scenes and conditions. Deep learning models can gradually improve their detection performance in these complex scenarios through continuous learning and updating. Thirdly, the eye feature extraction is integrated into the model training process without the need to manually construct feature engineering. Models can learn useful features directly from the original data, thus greatly improving development efficiency and model performance. Finally, deep learning models have higher complexity and capacity, and can represent more complex functions. It is conducive to solving problems that are difficult to handle with traditional machine learning. Deep learning models can further improve their performance and robustness by integrating several subnetworks or introducing attention mechanisms, *etc.*

Therefore, deep learning-based detection algorithms are gradually becoming a hotspot in the field of blink detection. Many scholars began to use deep learning to solve the problem of blink detection (*Dong, Wang & Abbas, 2021*). In summary, it is of great practical significance to review the research on blink detection based on deep learning. This review can provide a reference for researchers to apply the blink detection methods.

In this article, the research on blink detection algorithms based on deep learning is reviewed. Based on the current state of the application of blink detection, the six most commonly used deep learning models are described. In addition, the commonly used datasets for blink detection and the evaluation metrics of performance are compared and discussed. Furthermore, the challenges and limitations of deep learning-based blink detection methods are discussed. At last, possible research directions are provided for the development of blink detection.

The main contributions of this article can be summarized as follows:

1) This article is the first to exclusively focus on human eye blink detection, establishing a clear research scope.

2) We summarize six different deep learning models that can realize eye blink detection, and introduce the performance of these methods to help researchers understand the latest development trend.

3) This article introduces the most common benchmark datasets and evaluation metrics in the field of blink detection. Furthermore, the possible future directions in this field are discussed.

## Rationale and audience

Blinking, as a typical action of human body, indicates the current physiological and psychological state information of the individual. In recent years, people have gradually found that blink detection can be widely used in many fields of daily life. Therefore, a large number of works related to blink detection have been born in the past decade, and

satisfactory results have been achieved. The traditional computer vision method is the most commonly used method, but it suffers from poor generalization ability and robustness. Deep learning is one of the techniques that is gaining popularity and playing an important role in the field. The blink detection method based on deep learning has a powerful feature learning ability. Table 2 summarizes the applications of deep learning-based blink detection methods in various fields. The introduction of deep learning technology can better identify eye features and process eye movement sequences, and the detection accuracy is higher. Despite the significance and rapid progress of blink detection based on deep learning, there is a lack of such an exhaustive literature review. This literature review aims to fill this gap by conducting a comprehensive survey on blink detection in the context of deep learning.

Given that blink detection has applications in several domains, this review caters to a wide audience, including the human-computer interaction (HCI) field, the cybersecurity field, the medical field, and the transportation field. The target audience for this systematic review is human-computer interaction researchers, driver assistance systems engineers, medical and psychological researchers, and computer vision researchers. Computer vision researchers receive a quick overview of current results and methods in the field to advance their research. HCI developers can learn about the performance, advantages and disadvantages of different algorithms in order to choose the most suitable technical solution for their applications. Driver assistance system engineers can learn about new methods for fatigue state through this article. Medical and psychological researchers can learn about the trend of blink detection technology and explore the possibility of its application in clinical and experimental research through the review.

## SURVEY METHODOLOGY

In this study, we utilized the Preferred Reporting Items for Systematic Reviews and Meta-Analyses (PRISMA) standards. This approach ensures transparency, reproducibility, and systematization of the review process (Lu et al., 2024b).

### Research questions

The review has summarized and analyzed deep learning-based blink detection algorithms from 2017 to 2024. The following research questions (RQs) are thus formulated:

RQ1: What is the trend in the application of blink detection technology in recent years?

RQ2: What are the applications of blink detection technology in various fields?

RQ3: What are the different models applied in deep learning-based blink detection?

RQ4: What are the advantages and disadvantages of deep learning models applied to blink detection?

RQ5: What are the performance evaluation criteria for blink detection algorithms?

RQ6: What datasets are available for training and testing blink detection algorithms?

RQ7: What are the limitations and challenges of deep learning-based blink detection methods?

RQ8: What are the future research directions for blink detection?

**Table 2 Deep learning based blink detection methods in various fields.**

| Fields | Deep learning model | Methods | Accuracy/metrics | Datasets |
|---|---|---|---|---|
| Transportation | GP-BCNN (*Wang, Huang & Guo, 2019*) | Determining driver fatigue by detecting blinking frequency. | 97.10% | CEW |
| | Yolov7 (*Li et al., 2024*) | Determining fatigue by calculating the amount of time a driver spends with his eyes closed. | 99.00% | CEW |
| | Sequential NN (*Redhaei et al., 2022*) | Detecting driver drowsiness by calculating the number of eye closures in a sequence of eye states with a deep learning model. | 97.00% | National Tsing Hua University Computer Vision Lab's DDD (NTHUDDD) |
| | CNN_BILSTM (*Rajamohana et al., 2021*) | Euclidean distance of the eye is calculated with CNN and BILSTM to classify the features accordingly. Driver drowsiness is detected by monitoring the eye blink rate. | 90.67% | URL Mrl.Cs.Vsb.Cz/ |
| Security | LRCN (*Li, Chang & Lyu, 2018*) | Expose fake face videos generated with deep neural network models. | Auc = 0.99 | CEW |
| | CNN-LSTM-FCNs (*Saealal et al., 2022*) | Recognising ai-altered video by detecting whether it blinks or not in the video | 90.80% | Faceforensics++ |
| People with disabilities | Efficientnet (*Medeiros et al., 2022*) | Helping amyotrophic lateral sclerosis (ALS) by blinking as a signal for human-computer interaction | 86.97% | Eyeblink8 |
| | CNN (*Abiyev & Arslan, 2020*) | Helping spinal cord injuries control the mouse through head movements and blinking. | 97.42% | CEW |
| Medicine | LSTM (*Fogelton & Benesova, 2018*) | Detecting symptoms of incomplete blinking due to dry eye syndrome | F1-Score = 0.913 | Eyeblink8 |
| | LRCN (*Cruz et al., 2022*) | Determine the cause of dry eye by testing for complete and incomplete blinks. | F1-Score = 0.946 | Eyeblink8 |

## Study search and selection

Articles for this study were collected from a variety of databases, including IEEE Xplore, ACM Digital Library, Web of Sciences, ScienceDirect, and Google Scholar, covering high quality research articles in the field from 2017 to 2024. The keywords are derived from the proposed research questions and include terms such as "blink detection", "deep learning", "computer vision" and "eye state detection". Figure 2 shows the PRISMA analysis chart for articles selection. Table 3 shows the inclusion and exclusion criteria for the selection process. Among the 2,312 records searched, 48 articles were finally selected after screening.

Table 4 summarizes the number of articles published each year in the field of blink detection since 2016, and compares the total number of articles based on deep learning methods with other methods. From Table 4, it is clear that the percentage of the number of articles based on deep learning has gradually increased in recent years, indicating that this field is gradually becoming a focus of research.

## Eligibility criteria

Table 3 summarizes the inclusion and exclusion criteria for the studies. These criteria were used to determine the range of studies and to determine which studies should be included and which should be excluded. Inclusion criteria define the points that must be considered for the study. Exclusive criteria define which characteristics would cause the study to be

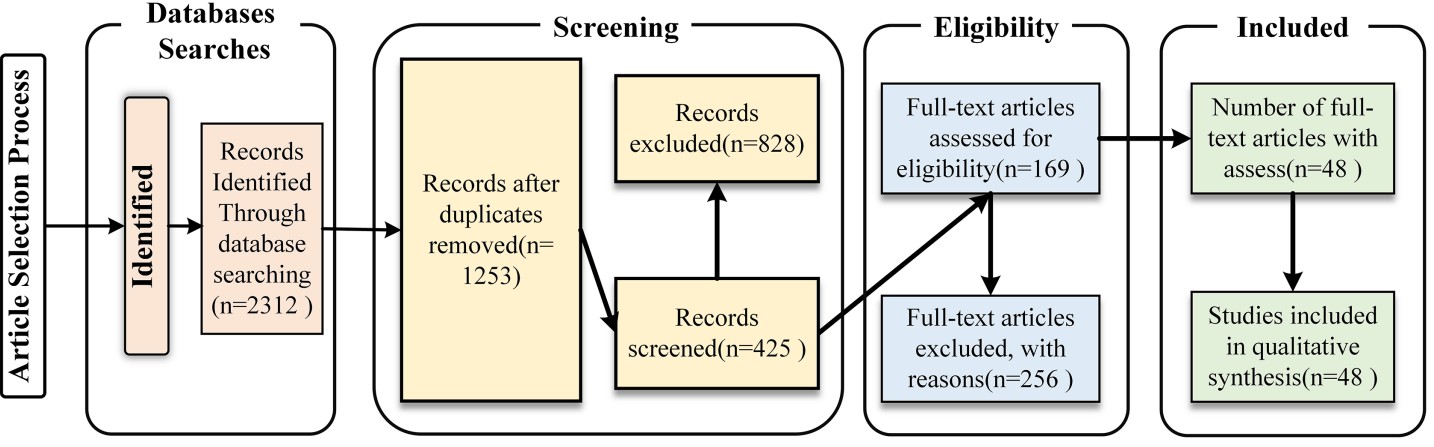

**Figure 2** PRISMA-based flowchart showing the studies selection process.

**Table 3 Inclusion and exclusion criteria used while searching the research article.**

| Inclusive criteria | Exclusive criteria |
|---|---|
| Studies that use deep learning algorithms as the primary method for detecting blinks. | Publications not related to blink detection. |
| Include the article between 2017 to 2024. | Removing the duplicates articles. |
| Include full-text articles. | Exclude the review and survey article. |
| Studies that are published are original articles in conference proceedings or journals. | Detection using non-deep learning methods. |

**Table 4 Summary of the number of articles per year.**

| Number of articles | 2017 | 2018 | 2019 | 2020 | 2021 | 2022 | 2023 | 2024 | Total |
|---|---|---|---|---|---|---|---|---|---|
| Deep learning based approach | 2 | 3 | 4 | 6 | 9 | 12 | 14 | 8 | 58 |
| Other methods | 31 | 38 | 34 | 44 | 61 | 57 | 48 | 33 | 343 |

excluded from the analysis. These criteria help to ensure that studies are focused and aligned with the study objectives.

## DEEP LEARNING ALGORITHMS IN BLINK DETECTION

After summarizing the approaches presented in this article, it can be observed that a majority of blink detection methods based on deep learning primarily rely on CNN, RNN, and transfer learning as their fundamental frameworks. Among them, CNN is an efficient and stable method in the field of image processing, which can accurately classify the eye states. RNN is mainly used to solve the sequence data problem, which can effectively deal with blinking video sequences. Transfer learning studies how to transfer the knowledge from the source domain to the target domain, which can save the resources and time for blink detection model training. In addition, a few researchers have applied Transformer and DBN algorithms to detect blinking. Consequently, this review predominantly focuses on investigating advancements made in these five domains.

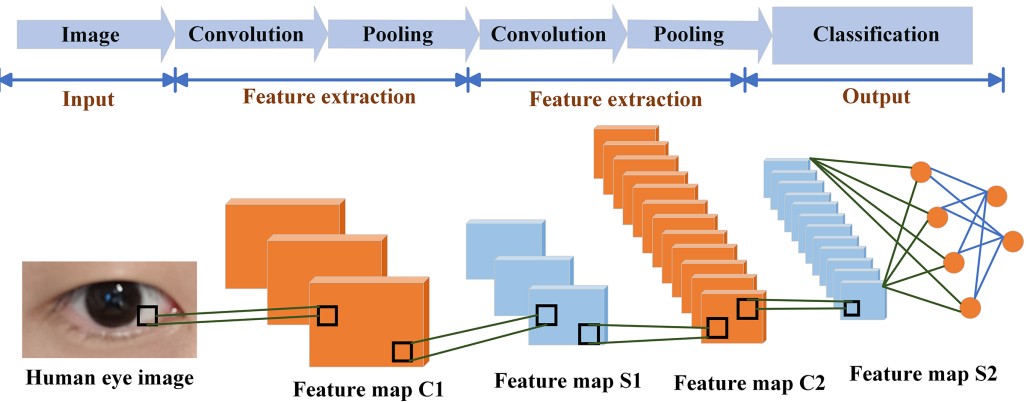

**Figure 3 Architecture of a convolutional neural network.**

## Application of convolutional neural network in blink detection

CNN (*Ma et al., 2021*) is widely used in the field of image processing. It uses the local characteristics of the data itself through spatial or temporal pooling, local perception and shared weights to achieve the effect of optimizing the network structure (*Fukushima, 1980*; *Krizhevsky, Sutskever & Hinton, 2012*; *Lecun et al., 1998*). Figure 3 shows that the blink detection model based on CNN usually consists of input layer, convolution layer, pooling layer and classification area (*Alzubaidi et al., 2021*; *Hussain, Bird & Faria, 2019*). The eye detection picture is fed from the input layer to the convolutional layer. The convolutional layer is an important part of the neural network, in which deeper abstract features are extracted (*Li et al., 2022*). The pooling layer contains two methods: average pooling and max pooling. The pooling layer can simplify the structure of the neural network.

The classical CNN models are AlexNet, VGGNet, GoogLeNet, ResNet. AlexNet (*Krizhevsky, Sutskever & Hinton, 2012*), proposed in 2012, is based on the LeNet-5 convolutional neural network structure and deepens the layers of the network. The model can deal with more complex image classification tasks and can also extract higher dimensional image features. A different feature from earlier neural network structures is that the ReLU activation function is used in AlexNet instead of the Sigmoid activation function to solve the problem of gradient dispersion. Dropout random suppression neuron correlation technique is also used to solve the overfitting problem. In 2014, *Simonyan & Zisserman (2014)* proposed VGGNet. One of the improvements of VGGNet over AlexNet is the use of several consecutive 3 × 3 convolutional kernels instead of the larger convolutional kernels in AlexNet (11 × 11, 7 × 7). For a given receptive field, the use of stacked small convolutional kernels is superior to the use of large convolutional kernels. And adding nonlinear layers allows the network to learn more complex patterns with fewer parameters. In 2015, *Szegedy et al. (2015)* proposed GoogLeNet, which has 22 layers, a further deepening from the 16 layers of VGGNet. Due to the increased depth and width of GoogLeNet, this network model is able to learn more and more detailed image features than previous deep learning models. GoogLeNet has fewer parameters than AlexNet and VGGNet due to the sparse network structure used in the model design and the proposed

Inception module. Traditional neural network models usually deepen the width and depth of the network structure to achieve better feature extraction. However, the network is usually easily degraded after deepening to a certain degree. In order to overcome this situation, *He et al. (2016)* proposed ResNet, and the shortcut structure of ResNet solves the depth degradation problem well.

During the blink detection, CNN continuously abstracts the data by convolution and pooling the extracted features. Local features are continuously refined into high-level features, and then the purpose of blink detection can be realized. In 2017, methods using deep learning to detect eye blinks gradually emerged. *Anas, Henríquez & Matuszewski (2017)* proposed two CNN architectures for detecting blinks based on LeNet: one of them is used to identify the eyes as two states (eyes closed and open), and the other for three-level eye state detection (eyes closed, eyes open, and partially eyes open). The experimental results show that the proposed model is superior to the model based on SVM and AdaBoost in the detection accuracy. It illustrates the effectiveness of CNN for blink detection to some extent. In order to further improve the accuracy of blink detection, *Sanyal & Chakrabarty (2019)* proposed a blink detection model based on two-stream convolutional neural network (TSCNN). The structure of TSCNN model is shown in Fig. 4, which consists of RGB-CNN and MCNN. RGB-CNN takes an RGB image of an eye as input, and MCNN takes a binary mask of the RGB image as input. The RGB-CNN network aims to learn the global features of the eyes. MCNN mainly learns the spatial distribution of eye shape, contour, eye region pixels and non-eye region pixels. Due to the different network structures of RGB-CNN and MCNN, they can extract different discriminative features of the eyes. Therefore, these two networks are combined and jointly trained to achieve the effect of eye state classification. The proposed model is more robust and accurate than its components RGB-CNN and MTCNN for blink detection alone. Experiments on various popular benchmark datasets show that the blink detector of the proposed model achieves 1–2% improvement in precision and recall. A robust, real-time, low-cost blink detection system was proposed by *Medeiros et al. (2022)*. A blink is used as a human-computer interaction signal, and CNN and SVM are used to handle the blink detection task. To start with, single shot multibox detector (SSD) is used for face detection. The eye region and its coordinates are extracted from the facial image by extracting the facial key points. Then CNN is used to classify the eye region to determine whether the eyes are open or closed, and SVM is used to classify the extracted eye coordinates. At last, based on the output of consecutive frames, the blink is detected and its duration is calculated. *Chavarro & Karakaya (2024)* proposed the use of blinks as an additional feature to address the poor performance of conventional iris recognition systems under non-ideal conditions. By using a pre-trained AlexNet model, the output layer was replaced with a regression layer, allowing the model to accurately output the percentage of blinks.

The blinking process has rich motion information. There are spatiotemporal variations at different scales. To this end, *Gao, Wang & Wang (2019)* proposed a two-level cascade model of video sequence-level preliminary detection and frame-level accurate detection based on CNN. Video clips with longer eye opening time and possible blinks were first separated. Then the blinking process is discriminated in the video clips where blinks may

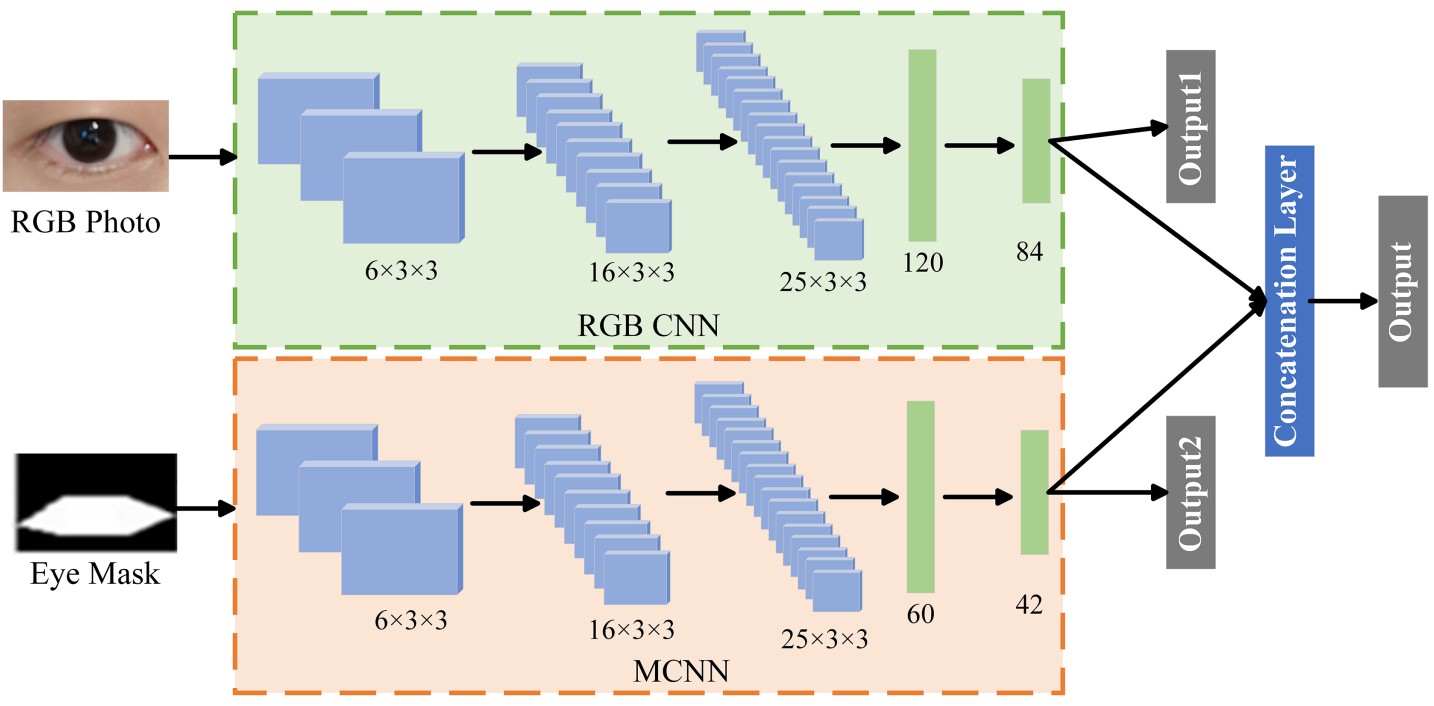

**Figure 4 Two-stream convolutional neural network architecture.**

exist. It solves the problem that CNN is difficult to analyze eye movement sequences. In 2022, *Bekhouche et al. (2022)* addressed the problem that previous learning methods could only deal with sequences containing only a single blink and ignored the case where multiple blinks were present. This article presents a fast framework for blink detection and blink verification. Multiple blinks are effectively extracted from the image sequence, and the eye signal for blink detection is extracted by constructing a feature-based matrix. The blink verification is then performed using Pyramidal Bottleneck Block Networks (PBBN). The pyramid bottleneck module is shown in the Fig. 5. The total number of modules in the architecture is reduced by using this module. Thus, the number of parameters is reduced and the inference time is shortened. This study provides a potential method for effectively detecting blinking in a variety of practical applications, including disabled communication, fake face detection, and face anti-deception.

In wild environments, blink detection methods are susceptible to adverse lighting conditions. *Hong, Kim & Park (2023)* designed a CNN based Blink Estimation With Domain Adversarial Training Network (BEAT network), which extracts features that are not affected by the domain. The method of domain adversarial training is used to solve the adaptability of deep neural network to poor lighting conditions and improve the detection system performance. *Ibnouf et al. (2023)* proposed a fatigue detection method based on blink detection for low light conditions. Firstly, the input image is preprocessed by applying a light enhancement algorithm to improve the brightness and contrast of the image, and then the eye region is extracted from the processed image using a Haar Cascade

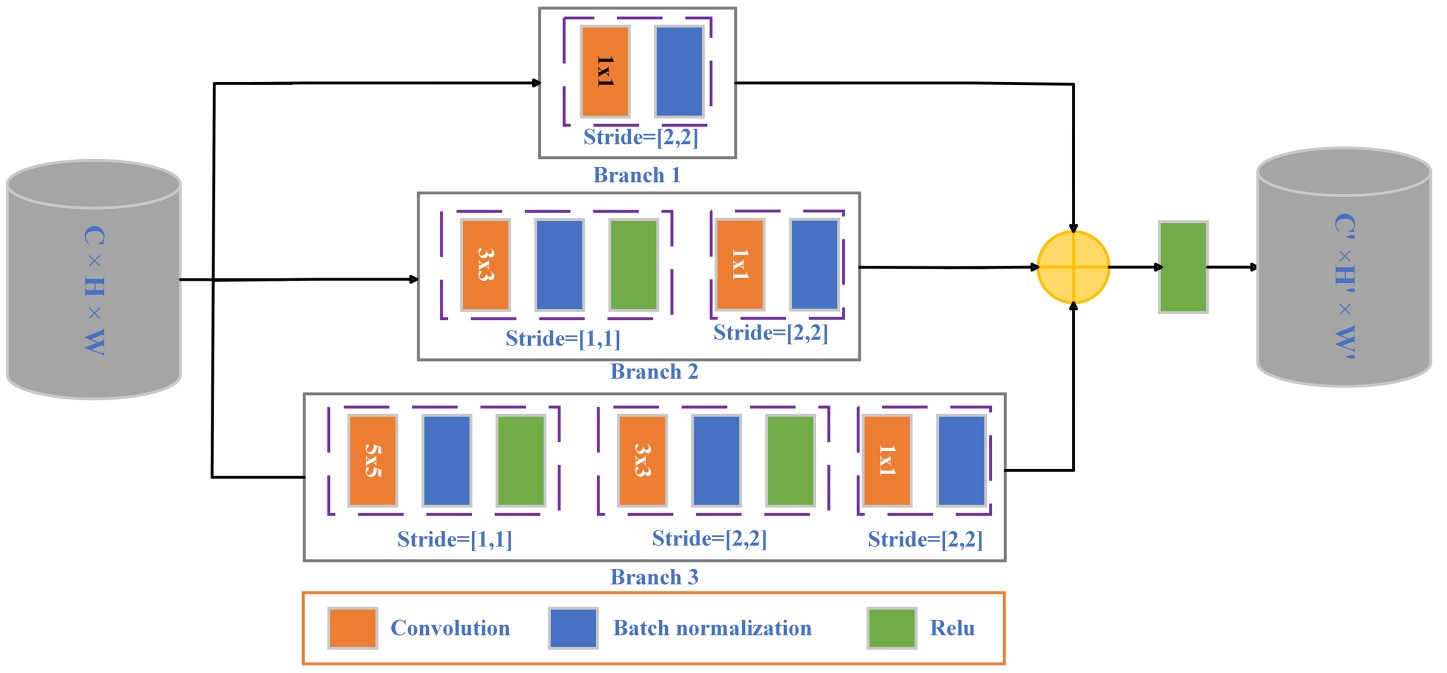

**Figure 5 Three-branch pyramid bottleneck block.**           

classifier. Next, the eye region images are input into a CNN model to determine whether the driver is in a fatigue state by calculating the blink frequency. *Zhou (2022)* adopted Zero-Reference Deep Curve Estimation (Zero-DCE) method based on deep learning to assist blink detection in view of the low accuracy of blink detection in low light environments. The main structure of Zero-DCE is seven convolution layers in symmetrical cascade, which is used to improve the details of dark and fuzzy images. This method can effectively improve the accuracy of blink detection results at night, and expand the application scenarios of detection. *Li et al. (2024)* proposed a YOLO-based eye state detection algorithm, referred to as ES-YOLO, to address the problems of complexity and susceptibility to light interference in blink detection algorithms. The algorithm optimizes the structure of YOLOv7, integrates multi-scale features using the convolutional block attention mechanism (CBAM), and improves the attention to the important spatial locations in the image. In addition, Focal-EIOU Loss is used instead of CIOU Loss to increase the attention to difficult samples and reduce the effect of sample category imbalance.

Table 5 summarizes the above CNN models. It can be obviously found that CNN has strong feature extraction ability. It extracts the features of the image step by step through the convolution layer and the pooling layer, so as to effectively represent the features of the eye region (*Cruz et al., 2022*). This feature extraction ability enables the CNN to capture key information such as the shape and texture of the eyes, which contributes to the accuracy of the blink detection task. CNN has a strong data adaptive ability. It can

**Table 5 Convolutional neural network summary on blink detection.**

| Model | Innovation | Advantages | Limitations | Precision | Recall | Data set | Detect time |
|---|---|---|---|---|---|---|---|
| LeNet (*Anas, Henríquez & Matuszewski, 2017*) | Convolutional neural network is introduced as a model for detecting eye blinks. | Real-time eye state detection can be achieved when head pose, appearance, and lighting conditions vary greatly. | The performance of the network strongly depends on the specific subset of data used for training. | 98.00% | 89.80% | ZJU | Nvidia GTX 960 GPU: 9.61 ms |
| TSCNN (*Sanyal & Chakrabarty, 2019*) | A two-stream convolutional neural network structure is proposed to jointly train RGB-CNN and MTCNN. | The blink detector achieves an improvement of 1-2% in ter ms of precision and recall. | Slow processing speed. | 98.85% | 99.12% | Eyeblink8 | 37.2 ms |
| CNN (*Gao, Wang & Wang, 2019*) | The spatiotemporal information of multiple channels such as single frame image, interframe optical flow, edge image and differential image is fused. The eye trends were described based on sparse optical flow between frames. | It solves the problem that CNN is difficult to analyze eye movement sequences. | Heavy computation. | 93.40% | — | ZJU | GeForce GTX TITAN Z GPU: 53.4 ms |
| Pyramidal—CNN (*Bekhouche et al., 2022*) | The pyramid bottleneck module is introduced to reduce the number of parameters. | Ability to handle multiple blinks. The detection speed is fast. | The required face image for the test must be the frontal image of the face. | 93.12% | 96.17% | Epan-EyeBlink | — |
| SSD and CNN (*Medeiros et al., 2022*) | Real-time detection with low delay is achieved through deeply separable convolution and CLAHE techniques. | The system is enabled to perform under real-time conditions, making it suitable for applications that require a fast response time; the required hardware costs are low. | — | F1-Scores = 0.924 | — | ZJU | i5-7200U process: 29.8 ms |
| BEAT Network (*Hong, Kim & Park, 2023*) | A domain adversarial training method for domain generalization is proposed, and a gradient decay layer is used to achieve stable adversarial training. | The proposed model can robustly detect eye blinks in images taken under weak lighting conditions by extracting domain-invariant features. | — | 95.77% | 80.22% | RT-BENE | RTX 2070Ti GPU: 3.4 ms |

(Continued)

| Model | Innovation | Advantages | Limitations | Precision | Recall | Data set | Detect time |
|-------|-----------|-----------|------------|-----------|--------|----------|-------------|
| CNN (*Ibnouf et al., 2023*) | Overcomes poor image quality in low-light conditions using light enhancement algorith ms | Works normally on low-quality images. | Multiple models need to be combined for detection and the detection process is complex. | Accuracy = 97.92% | — | Self-made datasets | — |
| AlexNet (*Chavarro & Karakaya, 2024*) | Blink detection algorithm is proposed as an additional feature in iris recognition system. | The model can handle images at different gaze angles and improves the recognition of non-ideal images. | The model requires high computational resources for training and inference and is not suitable for resource-constrained devices. | 90.00% | — | Off-angle iris dataset | — |
| YOLOv7 (*Li et al., 2024*) | Multi-scale features are integrated using CBAM increase attention to important spatial locations in the image. Furthermore, using Focal-EIOU Loss instead of CIOU Loss to increase attention to difficult samples. | The method has a small number of parameters, fast detection speed and easy access to hardware devices. | The accuracy of the algorithm decreases when the subject wears glasses or has a large angle of facial rotation. | 97.70% | 96.50% | CEW | NVIDIA GeForce GTX 1080Ti GPU: 7 ms |

automatically learn the patterns and rules in the blink detection task through large-scale training data, so as to improve the generalization performance of the model. In the case of experimental needs, CNN can be extended and improved by increasing the number of network layers, adjusting the network structure and parameters. To adapt to blink detection tasks of different scales and complexities (*Bekhouche et al., 2022*). However, CNNs have some drawbacks due to the complexity of CNN and the large number of parameters. The training and inference process for the blink detection task requires high computing resources and time, including GPU and memory (*Cortacero, Fischer & Demiris, 2019*). In addition, CNN is sensitive to illumination and pose changes when dealing with blink detection tasks. As a result, the model performs unstable under different illumination conditions or large changes in eye pose (*Zhao et al., 2018*).

In CNN-based blink detection research, a large amount of labeled data is required for training. Especially for blink detection tasks in complex scenes or specific application fields, more diverse training data are needed to improve the robustness of the model.

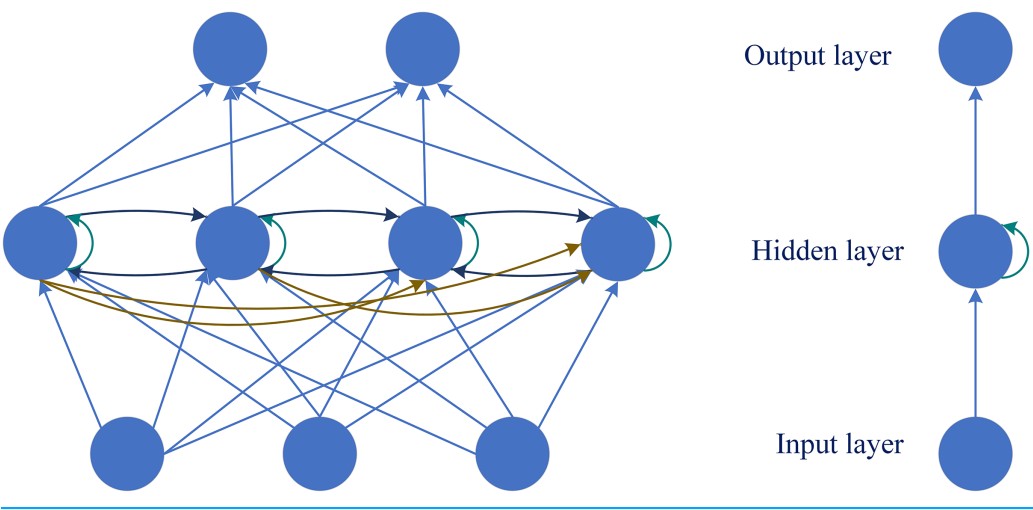

**Figure 6** Network structure of RNN.

However, in the blink detection task, the eye data in the normal state usually far exceeds the data in the blink state, which will lead to data imbalance (*Cortacero, Fischer & Demiris, 2019*; *Hong, Kim & Park, 2023*). The model is too biased towards the normal state to accurately detect blinking. This is a major difficulty in blink detection, and it is necessary to build a larger and more diverse blink data set in subsequent research. In order to provide better training samples and further improve the generalization ability and robustness of the model.

## Application of recurrent neural network in blink detection

RNN is a kind of self-connected neural network in the field of deep learning. It learns complex mapping from vector to vector and can be applied to various tasks related to time series (*Elman, 1990*; *Wang, Fan & Wang, 2021*). The model structure of RNN recurrent neural network is shown in Fig. 6 (*Tan et al., 2018*). The input of the hidden layer contains both the input of the current moment and the output of the hidden layer at the previous moment. The neurons in the hidden layer are connected to each other and can remember the previous information (*Dhruv & Naskar, 2020*; *Sherstinsky, 2020*). Therefore, it has advantages in dealing with timing problems in blink detection. The RNN-based blink detection models are described below and summarized in Table 6.

In blink detection, RNN plays the role of improving existing neural networks. Human perception of blinking is not just observing images, but viewing blinking as a behavioral process. It includes three stages: start, peak and end (*Gao, Wang & Wang, 2019*). *Fogelton & Benesova (2018)* used recurrent neural network as a classifier to detect eye blinks earlier. This is the first deep learning blink detection method capable of evaluating the integrity of blinks. By using RNN to detect the blink of the current frame based on the current and previous features, it provides a new idea for the research of blink detection.

In practical applications, training RNN will encounter the problems of gradient disappearance and gradient explosion, which makes it impossible to process very long eye video sequences (*Bengio, Simard & Frasconi, 1994*). *Hochreiter & Schmidhuber (1997)*

**Table 6 Recurrent neural network summary on blink detection.**

| Model | Innovation | Advantages | Limitations | F1-score/ metrics | Data set | Detect time |
|---|---|---|---|---|---|---|
| RNN (*Fogelton & Benesova, 2018*) | A recurrent neural network is introduced as a classifier to deal with sequence-based features. | Can distinguish between complete and incomplete blinks; Fast processing speed. | Blinks are misdetected when the person changes their head or eyeglasses position (movement within the eye area). | 0.913 | Eyeblink8 | — |
| MS-LSTM (*Hu et al., 2020*) | Concatenate the hidden layer states of the outputs of multiple LSTM units to contain richer timescale information. | Ability to take into account blinking behavior with different blinking durations. | Eye tracking is difficult when people move quickly. More discriminative blink behavior features need to be extracted. | 0.674 | HUST-LEBW (wild dataset) | TM i7-7700HQ CPU: 36.29 ms |
| LRCN (*Li, Chang & Lyu, 2018*) | The CNN and RNN models are combined to form the LRCN network | Strong specialty extraction ability; It can effectively handle temporal features. | Heavy computation | AUC = 0.99 | CEW | — |
| LRCN (*Cruz et al., 2022*) | The Siamese architecture is introduced in the CNN training phase. Transfer learning and data augmentation are used to cope with small training samples. | It overcomes the high-level imbalance problem in blink detection. Suitable for problems where blinking is defined as an action over time. | The working time of the feature extractor accounts for a large proportion of the total inference time of the model. | 0.946 | Eyeblink8 | NVIDIA Titan XP GPU: 111.7 ms |
| VGG16-LSTM (*Saealal et al., 2022*) | Applying the weight and bias updating process to adaptive moment estimation provides a way to gradually change the learning rate to adapt to changes in the dataset. | The model generalizes well across different datasets. | The method involves the use of multiple deep learning models in cascade, which increases the complexity of the model and requires higher computational resources and. | Accuracy = 90.80% | FaceForensic++ | — |
| InceptionV3-LSTM (*Popat et al., 2024*) | A two-stage network is introduced which combines eye features extracted *via* CNN with the detection of blink frequency by LSTM. | The method can be performed in real time on mobile devices to fulfil the need for real-time blink detection. | — | Accuracy = 91.40% | CEW | — |

proposed long short-term memory network (LSTM) in 1997 to achieve the purpose of improving the traditional recurrent neural network model. As shown in Fig. 7, an LSTM unit consists of a cell state $(C_t)$, an input gate $(\sigma_i)$, a forget gate $(\sigma_f)$, and an output gate $(\sigma_o)$. $\sigma_i$ and $\sigma_f$ work together to extract the information they need and filter out the noise. Information useful for recognizing the behavior is stored in $C_t$, which in turn updates the cell state. $\sigma_o$ is responsible for combining the current input and the valid information in the previous frame recorded in the hidden state $h_t$ to synthetically judge the output. After that, the hidden state is updated according to the current output and the updated cell state, which facilitates the utilization of subsequent LSTM units and enables it to cope with long-term dependencies in time series data (*Bahdanau, Cho & Bengio, 2015*; *Bhaskar & Thasleema, 2023*; *Houdt, Mosquera & Nápoles, 2020*). In 2020, *Hu et al. (2020)* targeted the multi-temporal nature of blinks. By extracting the multi-temporal scale features of the

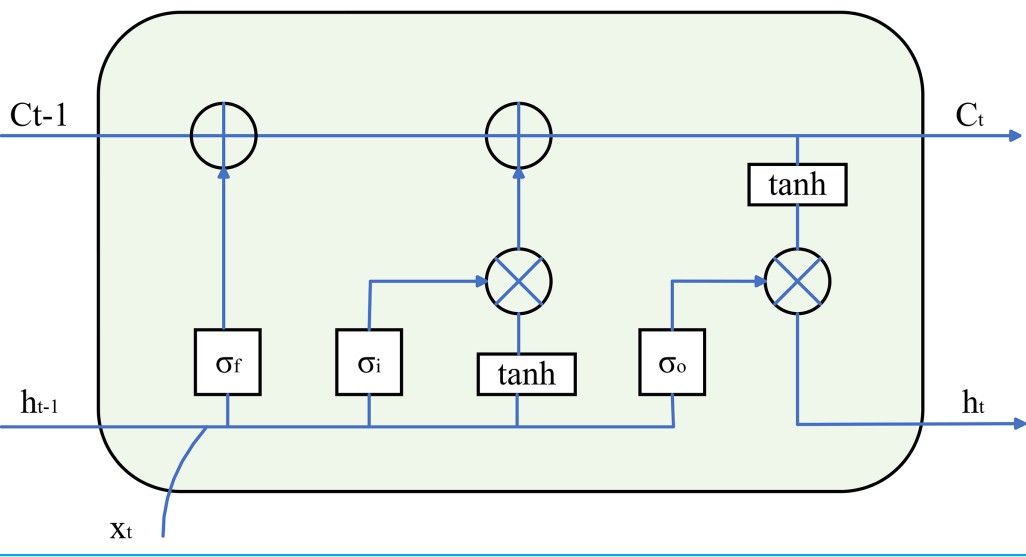

**Figure 7  Schematic diagram of the LSTM unit.**     

eyes, the modified LSTM model, Multi-scale LSTM (MS-LSTM), is used to discriminate the blink. It improves the adaptability of LSTM to detect blinks under unconstrained conditions.

The advantage of RNN in blink detection is that it can model the sequence data in a cyclic way and capture the time information of the blink action (*Fogelton & Benesova, 2018*). This ability allows the RNN to take into account the previous eye state in blink detection to better determine whether a blink is currently occurring. And because the blink action is sequential, RNN can effectively extract temporal features, such as the duration and frequency of blinks. It helps to distinguish more accurately between blinking and other similar actions (*Hu et al., 2020*; *Lu et al., 2024a*). However, RNN and LSTM are sequential. Each time step needs to be computed sequentially, resulting in low computational efficiency. In particular, it will bring some delay when dealing with longer sequences or real-time applications (*Cruz et al., 2022*).

### Application of CNN-RNN in blink detection

In engineering practice, there are few cases of applying RNN model alone for blink detection. Most researchers construct a fusion model of CNN and RNN to detect eye blinks. In 2018, *Li, Chang & Lyu (2018)* targeted CNN-based methods for their inability to consider temporal consistency during blinks (*Anas, Henríquez & Matuszewski, 2017*). For the first time, the combination model of RNN and CNN is used to discriminate eye blinks, and a long-term recurrent convolutional networks (LRCN) is formed, as shown in Fig. 8 typical CNN and RNN models. The CNN first identifies the discriminative features, and the output is passed to an RNN with long short-term memory for sequential learning of the temporal context of past and future inputs. Finally, the output of the RNN is classified as eye open or eye closed. It solves the problem that the state of the previous eye image cannot be taken into account. *Popat et al. (2024)* instead segmented the user's live video stream into a series of image inputs, trained a CNN to classify each frame independently to detect

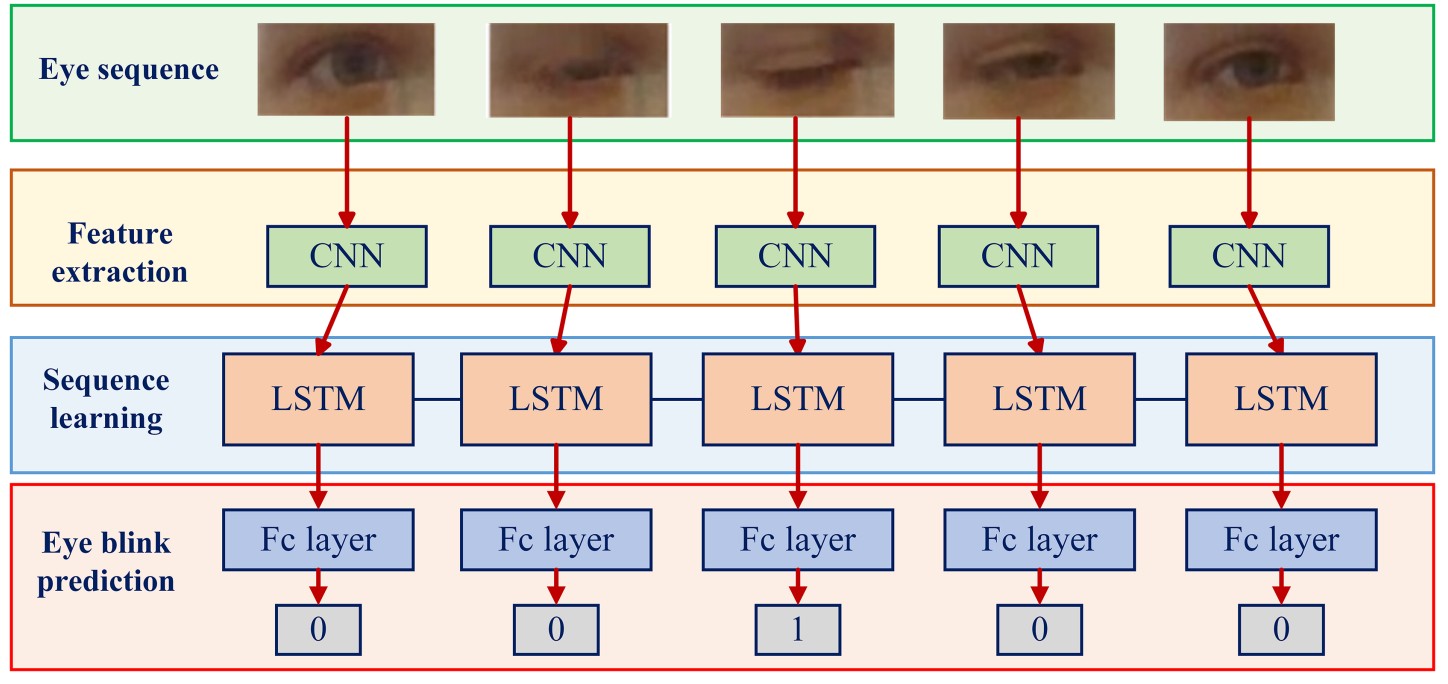

**Figure 8  Based on CNN and RNN fusion model structure diagram.**

blinks, and used a LSTM to calculate blink frequency. The method can be implemented as a background application on smartphones and computers to help prevent the risk of computer vision syndrome (CVS) or similar disorders. *Cruz et al. (2022)* and *Saealal et al. (2022)* improved the LRCN structure. Gonzalo et al. used the Siamese architecture during CNN training to overcome the high-level imbalance present in blink detection and the limited amount of data available to train the blink detection model. *Saealal et al. (2022)* applied the weight and bias update process to adaptive moment estimation (*Kinga & Adam, 2015*) to provide a direct way to gradually change the learning rate to adapt to changes in the dataset. Through these improved LRCN models, the performance is improved and the development of blink detection research is promoted.

It can be known from the above research on the blink detection method of CNN and RNN fusion. The biggest advantage of CNN is to extract eye image features, while RNN can effectively obtain the time information of the blink process. Therefore, many researches combine the advantages of the two methods to improve the classification effect of blink detection. When the actual eye area is small, the CNN-based model using only image input is ineffective, while the CNN-RNN model using temporal correlation can correctly predict. However, the model is composed of two deep learning networks, the structure is complex and the training is difficult. In some application scenarios, it is necessary to detect blinking accurately in real time, such as driver fatigue detection. However, RNN models usually require a long inference time (*Cruz et al., 2022*), which is not suitable for real-time applications. How to improve the inference speed of the model while maintaining the accuracy is a problem to be solved.

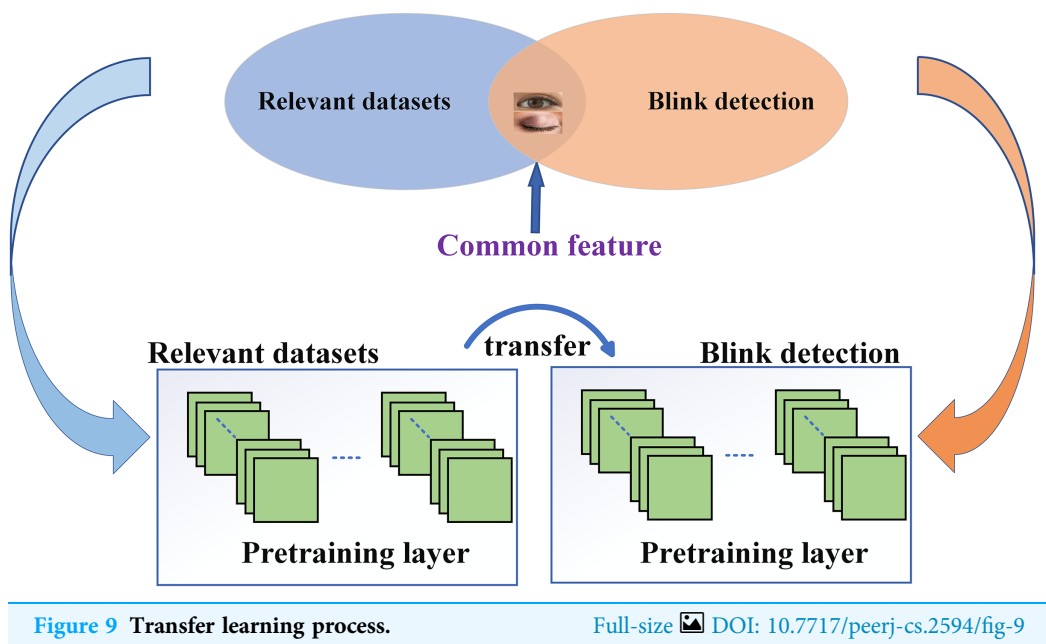

**Figure 9  Transfer learning process.**

## Application of transfer learning in blink detection

Transfer learning (TL) can be described as the transfer learning of knowledge learned in the source domain to the target task function (*Niu et al., 2020*; *Zhu et al., 2023*; *Zhuang et al., 2021*). In the field of blink detection, transfer learning can apply the prior knowledge learned from the blink data in the source domain to the target domain to accelerate the training of the model (*Kayadibi et al., 2022*). Figure 9 shows the transfer learning process for eye blink detection. Transfer learning is used to transform other neural network layers trained by blink-related databases into blink detection models (*Liu et al., 2021*).

The comparison of blink detection based on transfer learning and traditional machine learning is shown in Fig. 10. The figure shows the difference between the implementation process of traditional machine learning techniques and transfer learning techniques: traditional machine learning methods train different models independently for specific domains, data and tasks (*Tan et al., 2018*). Transfer learning can effectively use the existing knowledge from one domain to help improve the learning ability of other different target domains. In particular, it is used to overcome the limitations of highly specialized but small datasets (*Weiss, Khoshgoftaar & Wang, 2016*). Moreover, transfer learning can reuse the knowledge of previous tasks without having to learn a new task from the initial stage.

Blink detection using deep learning methods requires a large number of samples to train. It is difficult to obtain a sufficient number of datasets to meet the training requirements. In response to this problem, *Zhao et al. (2018)* used transfer learning methods to improve deep ensemble neural networks for blink detection. The base model is pre-trained by learning from the base dataset with enough samples through supervised learning. Then, the learning parameters of the pre-trained base model are transferred to the target network to fine-tune the data set of the target task. The recognition ability of

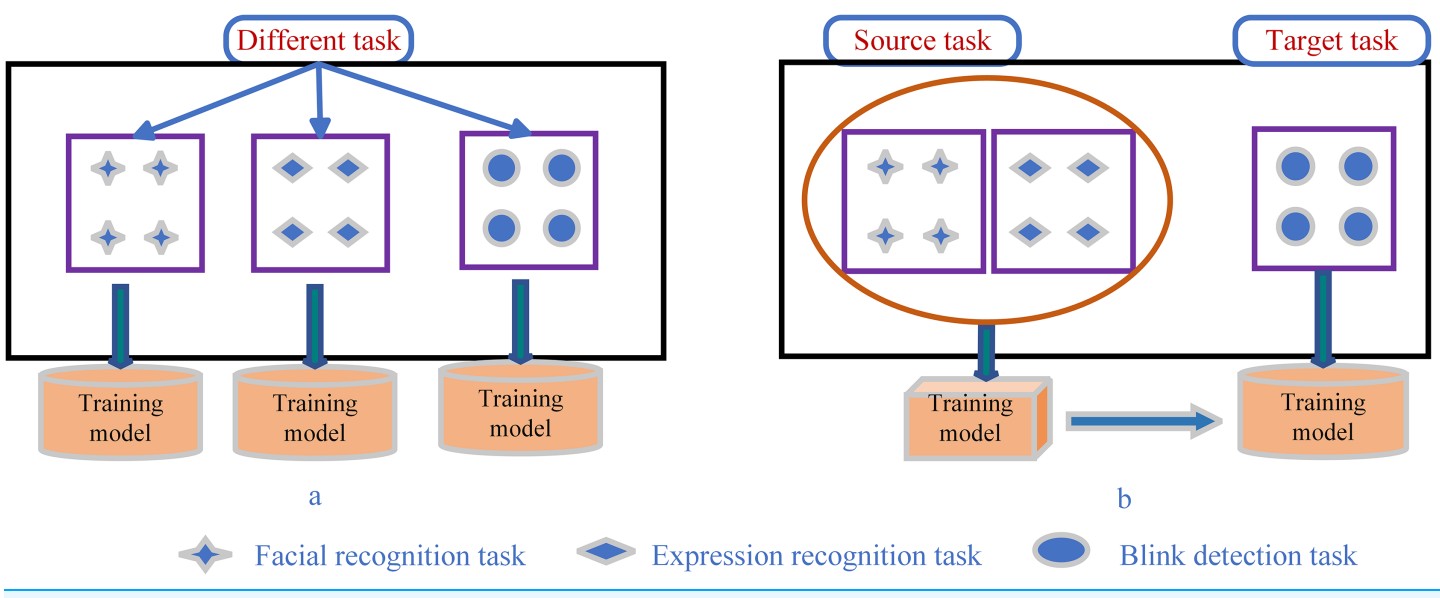

**Figure 10 Differences between (A) traditional machine learning and (B) transfer learning.**

deep learning method is improved. *Cruz et al. (2022)* also used transfer learning to process a small number of samples in their blink detection network for the problem of relatively small data sets. Making its method robust to class imbalance. In 2021, *Liu et al. (2021)* pointed out that the facial expression recognition task and the eye state detection task have a common cross feature of the eye region. A transfer learning method is proposed to accelerate the training speed of eye state detection method and improve the performance of the detection model. The introduction of transfer learning improves the parameter training in the network and reduces the limitation caused by the small number of training samples. In 2022, *Saurav et al. (2022)* addressed the issue that training a deep learning model on a small blink detection dataset may lead to overfitting. The transfer learning method is used to fine-tune the network parameters to overcome overfitting problem. *Pahariya, Vats & Suchitra (2024)* proposed a migration learning based MobileNetV2. Existing image classification datasets are used to train MobileNetV2, and then the model is adapted to the blink detection task by migration learning. This approach effectively improves the generalization ability and computational efficiency of the model. Table 7 summarizes the above transfer learning-based blink detection methods.

The advantage of the application of transfer learning in blink detection is the ability to transfer the learned weights to other neural network layers. The process of re-training the neural network is avoided and the training efficiency is improved (*Liu et al., 2021*). Transfer learning uses the learned features and knowledge to optimize the network parameters so as to improve the performance of the model and make up for the shortage of samples, and improve the performance of the model (*Kayadibi et al., 2022*). However, the application of transfer learning is limited by the similarity between the source and target

**Table 7 Transfer learning summary on blink detection.**

| Method | Year | Innovation | Accuracy/ metrics | Data set | Detect time |
|---|---|---|---|---|---|
| DINN (*Zhao et al., 2018*) | 2018 | With the help of knowledge learned from the facial expression dataset transferred to blink detection, the generalization ability and classification accuracy of the model on small sample datasets are improved. | 97.20% | ZJU | — |
| CNN (*Liu et al., 2021*) | 2021 | Transfer learning based utilizes common features of facial expression recognition task and eye state detection task to speed up training. | 97. 40% | CEW | — |
| CNN (*Saurav et al., 2022*) | 2022 | A transfer learning approach to overcoming the overfitting problem caused by small amount of training data by fine-adjusting the network parameters. | 97.59% | CEW | Nvidia Xavier: 16.1ms |
| ResNet18-LSTM (*Cruz et al., 2022*) | 2022 | Improving eye data category imbalance through transfer learning. | F1-score = 0.946 | Eyeblink8 | NVIDIA Titan XP GPU: 111.7ms |
| MobileNetV2 (*Pahariya, Vats & Suchitra, 2024*) | 2024 | Adaptation of pre-trained models through transfer learning to improve the generalization ability and computational efficiency of the models. | 86.00% | MRL eye dataset | — |

domains. If the two domains are very different, transfer learning will not bring significant performance improvement (*Saurav et al., 2022*).

## Application of transformer in blink detection

Transformer was first published by the Google research team in 2017 (*Vaswani et al., 2017*). The model uses multi-head self-attention (MSA) mechanism and Positional Encodings (PE), which can efficiently parallelize computation and capture long-range dependencies. Additionally, it demonstrates strong scalability and generalization capabilities. As a result, Transformer has established itself as a pivotal framework in the field of deep learning.

In the work published by *Liu, Xu & Lu (2023)*, a Transformer-based blink detection algorithm is proposed. Transformer architecture effectively addresses the challenges related to temporal feature extraction and parallelism in RNN-based methods. In addition, the algorithm extends an additional head to detect the blink strength in each frame. This enables more accurate learning of the key features of blinks. *Fodor, Fenech & Lőrincz (2023)* proposed a fast transformer-based blink detection method to address challenges such as lighting changes and head pose changes. The model fuses multiple input features through a multimodal Transformer, effectively combining low and high level feature sequences to achieve accurate blink detection and eye state recognition. *Hong et al. (2024)* proposed a Dual Embedding Video Vision Transformer (DE-ViViT) for blink detection. When a facial feature is detected in a frame, both eye regions are cropped according to their coordinates. Every 10 frames are passed to the dual embedding module, which consists of tubelet embedding and residual embedding. After converting the input sequences into embedding vectors, they are processed through the Transformer

**Table 8 Transformer and DBN summary on blink detection.**

| Model | Innovation | Advantages | Limitations | F1-score | Data set |
|---|---|---|---|---|---|
| BlinkLinMulT (*Fodor, Fenech & Lőrincz, 2023*) | An improved Linear Attention Multi-Modal Transformer (LinMulT) is proposed, which enables the model to better capture the interrelationships between different features by taking into account a variety of inputs, such as RGB texture, iris, ear, and head pose angles. | The ability of the model to recognize complex eye states was improved by combining low-level (*e.g.*, RGB texture) and high-level (*e.g.*, distance between eyelid marker points) feature sequences. | Performance will be affected when dealing with extreme head postures such as head down or head up. | 0.991 | EyeBlink8 |
| BlinkFormer (*Liu, Xu & Lu, 2023*) | A multi-task learning method for detecting blinking behavior and strength at the same time is proposed, which improves the generalization ability of the model in complex scenarios. | It is highly generalizable and can be applied to other similar time series analysis tasks. | The model is computationally expensive to train and inference, so it is not suitable for resource-constrained environments. | 0.843 | HUST-LEBW |
| DE-ViViT (*Hong et al., 2024*) | Two different embedding strategies were used to capture significant and small changes in the eye movement sequences. | Combining the MSA mechanism with MLP blocks for feature transformation and classification prediction to improve the accuracy and robustness of the model. | A two-stage strategy was used, where eye detection was performed before recognizing blinks, limiting the speed of detection. | 0.853 | HUST-LEBW |
| DBN (*Kir Savaş & Becerkli, 2022*) | The RBM in DBN is able to extract complex features layer by layer to capture the subtle changes in blinking movements. | It is can keep good performance in the case of limited training data. | The training process involves the training of multiple RBM layers, which requires a lot of computational resources and time. | 0.713 | NthuDDD |

architecture. Finally, multilayer perceptron (MLP) head detects if a blink occurs based on the output of the encoder.

The blink detection model based on Transformer shows strong multi-task learning capability and feature capture ability, which can accurately recognize complex eye states and improve the robustness of detection. However, Transformer requires substantial computational resources and is not suitable to resource-constrained environments. Furthermore, the self-attention mechanism in the Transformer model is based on global information, which is less effective in detecting small targets (*e.g.*, small changes when eyes are closed).

## Application of deep belief network in blink detection

In 2006, *Hinton & Salakhutdinov (2006)* utilized the stacked dimensionality reduction technique of multi-layer restricted Boltzmann machines (RBM) to achieve an unsupervised feature learning approach, but the problems of unstable gradient in training and sensitivity to raw data were not solved. In the same year, *Hinton, Osindero & Teh (2006)* added greedy algorithms and linear classifiers to form deep belief networks (DBN), which is widely used nowadays. In 2022, *Kir Savaş & Becerkli (2022)* used the method of

**Table 9 Comparison of advantages and disadvantages of blink detection methods based on deep learning.**

| Model | Advantages | Disadvantages |
|---|---|---|
| CNN | Strong feature extraction ability. Strong data adaptive ability. It can be extended and improved by increasing the number of network layers and adjusting the network structure and parameters. | It is sensitive to illumination and pose changes. A large amount of labeled data is required for training. The structure is complex when applied to process the blink sequence. |
| RNN | The time information of the blink action can be obtained. The temporal features can be extracted effectively. | It will introduce some delay when processing longer sequences or real-time applications. It is difficult to extract the features of eye images. |
| CNN-RNN | It can extract eye features and time information simultaneously. | The structure is complex and the training is difficult. |
| Transfer learning | No retraining of the neural network. It reduces the limitation of insufficient training samples and improves the parameter training in the network. | There should be some similarity between the source and target domains. |
| Transformer | It can deal with multi-scale blink detection tasks by performing feature fusion at different scales. It is able to capture global contextual information in images through a self-attentive mechanism. | The computational complexity is high. The detection speed cannot reach real-time. |
| DBN | Unsupervised pre-training in DBN can learn the underlying features of the data without labeling. | Both the training and inference processes require significant computational resources. |

detecting drivers' blinking behavior as an indicator for assessing driving fatigue by using DBN. RBM automatically extracts features at different abstraction levels through unsupervised learning. After layer-by-layer pre-training, the DBN is subjected to supervised fine-tuning using labeled data to optimize its recognition of blinking. Experiments show that the model can effectively detect the blinking behavior of drivers. Table 8 summarizes the above Transformer and DBN based blink detection methods.

The DBN-based blink detection algorithm maintains good performance despite insufficient data, which is attributed to the capability of extracting hierarchical features through unsupervised learning to provide rich and deep data representations for subsequent blink detection. However, the training process of this algorithm involves the training of multiple RBM layers, which will cost a lot of time.

Based on the analysis presented above, a comparison of the advantages and disadvantages of deep learning methods, including CNN, RNN, transfer learning, Transformer and DBN in the context of blink detection applications, is provided in Table 9. This comparison aims to facilitate a more effective utilization of these methodologies for advancing research in blink detection.

# COMMON DATASETS AND EVALUATION METRICS

## Commonly used datasets

Datasets play a fundamental role in blink detection and are an important factor affecting the development of this research field. Currently, the commonly used datasets in eye blink detection tasks are mainly Closed Eyes in the Wild (CEW) (*Song et al., 2014*), ZJU (*Pan et al., 2007*), Eyeblink8 (*Drutarovsky & Fogelton, 2015*), Constraint-plus (*Ghoddoosian, Galib & Athitsos, 2019*), Helen (*Le et al., 2012*), and Real-Time Blink Estimation in Natural

**Table 10 Common datasets for blink detection.**

| Dataset name | Resolution | Type of dataset | Number of samples | Access | Number of papers used | Source link |
|---|---|---|---|---|---|---|
| CEW | Various | Image | 2,423 | Open | 65 | https://parnec.nuaa.edu.cn/_upload/tpl/02/db/731/template731/pages/xtan/ClosedEyeDatabases.html |
| RT-BENE | 1,920 × 1,080 | Image | 200,000 | Open | 13 | https://zenodo.org/records/3685316 |
| Constraint-plus | Various | Video | 4,935 | Open | — | sites.google.com/view/utarldd/home |
| Helen | 400 × 400 | Image | 2,330 | Open | 188 | https://exposing.ai/helen/ |
| ZJU | 320 × 240 | Video | 8,984 | Open | 201 | http://www.cs.zju.edu.cn/ |
| Eyeblink8 | 640 × 480 | Video | 70,992 | Open | 32 | https://www.blinkingmatters.com/research |

Environments (RT-BENE) (*Cortacero, Fischer & Demiris, 2019*). Table 10 organizes the above commonly used datasets for eye blink detection. The emergence of these datasets has promoted the development of deep learning techniques in eye blink detection research.

CEW: The CEW dataset is extracted from LFW dataset (*Huang et al., 2008*). The dataset has a total of 2,423 images, of which 1,192 images with closed eyes are collected directly from the Internet and 1,231 images with open eyes are selected from labeled faces in the field database. This dataset involves a wide range of people, and various human eye pictures are comprehensive. But the images provided by each subject lacked variety.

RT-BENE: The RT-BENE dataset has annotations of eye open degree of more than 200,000 eye images. Includes over 10,000 images of eyes closed. This dataset is suitable for gaze estimation in natural environments where the distance between the camera and the subject is large and the subject motion is less constrained. Its large size is an important benchmark in blink detection research.

Constraint-plus: Constraint-plus is established based on the time-series blink behavior dataset RLDD. It combines existing constrained blink datasets (ZJU, Eyeblink8, Talking-faceRLDD, *etc.*) into a new constrained blink dataset. The dataset contains 4,935 samples (2,435 blink samples and 2,500 non-blink samples). Among them, the training set contains 2,235 blink samples and 2,300 non-blink samples, while the test set contains 200 blink samples and 200 non-blink samples. This dataset is able to illustrate whether the superimposed constrained dataset is a good description of the blinking behavior under unconstrained conditions.

Helen: The Helen dataset contains a large number of images of objects with open and half-open eyes. However, the number of subjects with closed eyes is rather limited. There are 2,330 images in the Helen dataset, consisting of facial images representing different ages, genders, and ethnicities. And the main face locations are accurately and detailed annotated. This dataset contains images with different poses, illuminations, and expressions, and is suitable for training models that extract facial features from the wild.

ZJU: The ZJU dataset consists of 80 videos of 20 people. Clips of several seconds were collected with a 30fps camera with a resolution of 320 × 240. Each person has four segments, which are the frontal view, the upward view, the view with glasses, and the view

without glasses. Facial expressions and head movements are absent from the collected segments. Some of these images have low resolution or are occluded by glasses.

Eyeblink8: The Eyeblink8 dataset is a dataset of eye blinking behavior. Eight videos with different temporal lengths are included. Every two videos belong to the same person (one of them is wearing glasses). The entire dataset contains 70,992 images with a resolution of 640 by 480. Videos were recorded under different conditions, with most faces facing directly towards the camera. It contains many natural facial movements and other non-blinking actions, increasing the challenge of blinking detection.

## Evaluation metrics

In order to compare different blink detection algorithm models, evaluation metrics are needed. The evaluation metrics include precision, recall, F1-score, accuracy *etc*. Their numerical values are obtained from the variables TP, TN, FN, FP in the confusion metrix (*Ahmed et al., 2023*). Where TP is the number of blink samples that were correctly identified. TN represents the number of non-blinking samples correctly identified. FN represents the number of blinking behaviors that were detected as non-blinking behaviors. FP represents the number of non-blink samples detected as blink samples.

The blink detection precision (P) is defined as follows:

$$P = \frac{TP}{TP + FP} \tag{1}$$

Recall (R) is defined as follows:

$$R = \frac{TP}{TP + FN} \tag{2}$$

Accuracy is defined as follows:

$$Accuracy = \frac{TP + TN}{TP + FP + TN + FN} \tag{3}$$

The F1-score of blink detection is defined as the harmonic average of accuracy and recall. Its formula is as follows (*Cruz et al., 2022*):

$$F1\text{-score} = 2 \times \frac{P \times R}{P + R} \tag{4}$$

In addition, ROC-AUC is also an important metric for evaluating the performance of blink detection models. The horizontal coordinate of the receiver operating characteristic (ROC) curve is FPR, *i.e.*, how many samples that are actually non-blinking are predicted to be blinking. The vertical coordinate is TPR, *i.e.*, how many of the actual blinking samples are predicted to be blinking. Area under curve (AUC) is the area under the ROC curve, and its value can evaluate the model. The larger the value, the better the model classification performance.

## CHALLENGES AND LIMITATIONS

Significant progress has been made in deep learning-based blink detection, but the field still has untapped potential and faces various challenges. These challenges include the following:

- **Imbalance in the amount of data in different categories:** In the blink detection dataset, the data of non-blinking states far exceeds that of blinking states, leading to an imbalance in the amount of data for eye states (*Cortacero, Fischer & Demiris, 2019*; *Hong, Kim & Park, 2023*). This causes the model to be ineffective in detecting the blink state after training, while failing to accurately detect blinks. Therefore, how to deal with the data imbalance of eye states in the dataset is a challenge that needs to be addressed in the model training phase.

- **The effect of light changes on blink detection:** Blink image acquisition is mainly focused on indoor environments, and this method can effectively eliminate external light interference and simplify the image processing process. When detecting blinks outdoors, the changes in natural light are very dynamic. The captured video may appear to be too bright or too dark, which reduces the discriminability of eye features and poses a great challenge for accurate recognition.

- **Impact of complex situations:** Most blink detection studies have focused on simple backgrounds and have not considered the effects of complex environmental disturbances such as occlusion and posture changes (*Hong, Kim & Park, 2023*; *Zhao et al., 2018*). Occlusion and posture changes are common in reality. Occlusion includes occlusion of eyes by glasses and hair *vs.* light occlusion caused by illumination. Pose changes include head rotation and body rotation. All of these complex conditions can reduce the recognizability of eye features, leading to false and missed detection problems. As a result, the accuracy of blink detection under complex conditions is low and practical applications are greatly limited.

- **The problem of real time:** Applying deep learning models to practical blink detection, real-time detection is a key requirement. Real-time detection requires that the detection speed can be taken into account while ensuring the detection accuracy. Deep learning-based blink detection models have high detection accuracy, but they have a large number of parameters and require a long inference time (*Cruz et al., 2022*). Therefore, a challenge for blink detection is how to increase the model's inference speed and lighten the model while maintaining accuracy.

- **Limitations of application equipment:** Deep learning models have a high dependence on hardware devices, and the detection effect obtained will be better when computed on GPUs with fast computing speed and high computing accuracy. However, blink detection technology is usually applied in mobile devices and edge devices in practice, and the computational efficiency of these devices is low, which cannot meet the demand of deep learning algorithms for high-performance computing. In addition, the deep learning models need to consume a large memory cost, and the application on hardware devices with limited memory resources will be limited. Therefore, how to overcome the

dependence on hardware devices so that deep learning models can achieve better experimental results even on ordinary devices is a challenge for the future development of blink detection.

## FUTURE DIRECTIONS

Based on the current development status of blink detection, and the challenges it faces, this article looks into the future research directions in the field of blink detection as follows:

- **Reducing model complexity:** deep learning-based blink detection methods have superior performance compared to traditional manual feature methods due to their powerful feature extraction capabilities, but at the cost of requiring more computation and storage. In order to improve the real-time performance of deep learning-based blink detection algorithms and enhance their grounded applications. Researchers can design lightweight deep learning models, such as MobileNet (*Sandler et al., 2018*), SqueezeNet (*Yin et al., 2024*), *etc.*, which reduce the number of parameters and computation while maintaining high accuracy and improve the inference speed. In addition, simple network structures can be redesigned by methods such as model pruning and knowledge distillation to further reduce the model size.

- **Multimodal fusion:** combining multimodal data obtained from different sensors such as EOG signals, infrared images, facial expressions, *etc.*, to realize a multimodal fusion blink detection system. This provides more comprehensive information and further improves the accuracy and reliability of detection.

- **Complex detection scene facing:** blink detection will appear in complex detection scenes that affect the detection results. For the impact of lighting changes, the application of data enhancement algorithms should be considered. Adjust the local brightness value of the image to enhance the image detail effect and reduce the impact of lighting factors. For the impact of complex situations, more robust feature extraction and model training methods can be developed, such as using attention mechanisms to focus on the eye region to reduce the impact of occlusion and pose changes on the detection results. Also, 3D modeling based methods can be researched to deal with problems such as head rotation.

- **Developing hardware devices suitable for deep learning:** currently deep learning models can achieve good performance on high-performance GPUs. However, in practical applications mobile devices and edge devices usually cannot meet the demand of deep learning models for high-performance computing. Therefore, the development of hardware devices suitable for the operation of deep learning algorithms is a worthwhile research problem. Such hardware needs to show speed advantages in image data processing, neural network inference, *etc.*, and possess the advantages of low cost, low power consumption, and programmability. The successful development of such hardware will make it possible for deep learning-based blink detection to be widely used in mobile and edge devices.

# CONCLUSION

Deep learning can extract complex features to identify and classify and solve problems. It has achieved good application results in the field of blink detection. In this article, we review the application of deep learning in the field of blink detection, analyze the common model architecture of deep learning applied to blink detection. The summary for deep learning models helps researchers to understand the characteristics of these methods and provides them with a reference for choosing appropriate deep learning models. This article provides a summary of datasets and evaluation criteria applied to blink detection. Information on blink detection-related datasets is listed in detail, researchers can apply these datasets to construct their own deep learning models. Finally, this article discusses the challenges and research directions of deep learning in the field of blink detection, providing valuable insights for researchers to promote progress and innovation in the field.

## Funding

This work was supported by the National Natural Science Foundation of China—Guangdong Province Joint fund key project (No. U22A20221), National Natural Science Foundation project (No. 62073090), Guangdong Province university key field special (No. 2020ZDZX2014) and Special Fund for Scientific and Technological Innovation Strategy of Guangdong Province under Grant (No. PDJH2024A225). The funders had no role in study design, data collection and analysis, decision to publish, or preparation of the manuscript.

## Grant Disclosures

The following grant information was disclosed by the authors:
National Natural Science Foundation of China—Guangdong Province Joint fund key project: U22A20221.
National Natural Science Foundation project: 62073090.
Guangdong Province university key field special: 2020ZDZX2014.
Scientific and Technological Innovation Strategy of Guangdong Province: PDJH2024A225.

## Competing Interests

The authors declare that they have no competing interests.

## Author Contributions

- Jianbin Xiong conceived and designed the experiments, performed the experiments, prepared figures and/or tables, authored or reviewed drafts of the article, and approved the final draft.
- Weikun Dai conceived and designed the experiments, performed the experiments, performed the computation work, prepared figures and/or tables, authored or reviewed drafts of the article, and approved the final draft.

- Qi Wang conceived and designed the experiments, performed the experiments, prepared figures and/or tables, authored or reviewed drafts of the article, and approved the final draft.
- Xiangjun Dong analyzed the data, prepared figures and/or tables, authored or reviewed drafts of the article, and approved the final draft.
- Baoyu Ye analyzed the data, prepared figures and/or tables, and approved the final draft.
- Jianxiang Yang analyzed the data, performed the computation work, prepared figures and/or tables, and approved the final draft.

## Data Availability

This is a literature review.

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
