# Peer review of "A review of deep learning in blink detection"

_PeerJ Computer Science, doi:10.7717/peerj-cs.2594_

## Round 0.1 · original submission · Major Revisions

· Academic Editor

Major Revisions

Dear authors,

Thank you for submitting your Literature Review article. Feedback from the reviewers is now available. It is not recommended that your article be published in its current format. However, we strongly recommend that you address the issues raised by the reviewers and resubmit your paper after making the necessary changes. Before submitting the paper following should also be addressed:

1. Please provide a clearly defined research question for this literature review paper;
2. Clearly reported, reproducible, and systematic methods should be provided in order to identify, select, and critically appraise relevant research.
3. The Abstract should be attractive and contain motivation.
4. The Introduction section should adequately introduce the subject and make it clear who the audience is and what the motivation is.
5. The future research directions should be itemized and discussed in more detail.

Best wishes,

Reviewer 1 ·

Basic reporting

The article do not contain any block diagrams , A Prisma chart can be included un the literature review at the beginning. More over a basic block diagram is needed. Writing can be improved

Experimental design

Article is within the aim and scope. Blink detection can be from various modalities' such as facial images, videos, EEG or EOG. A comparative table showing the literatures with various modalities should be included.

Validity of the findings

Some Future directions in terms of advancements in deep learning and signal processing can also be included.

Separate headings on pre processing and signal processing techniques for feature extraction is appreciable.

Additional comments

Comparative tables with applications are expected. Also a small example of blink detection to be included as analysis.

Cite this review as
Anonymous Reviewer (2025) Peer Review #1 of "A review of deep learning in blink detection (v0.1)". PeerJ Computer Science

Reviewer 2 ·

Basic reporting

The manuscript has been great, but still need to be improved.
1. To increase reader interest, the abstract section should briefly explain the research results.
2. In the introduction, explain the advantages and disadvantages of deep learning in detail (lines 90-98)?
3. The traditional computer vision method is the most commonly used method, but it is difficult to find an optimal feature extraction method and feature dimension to extract features as the input of the blink detection classifier. describe more details about traditional computer vision

Experimental design

Already good

Validity of the findings

Already good

Additional comments

1. About 80 scientific papers were found to be relevant to the aforementioned index terms and about 40 scientific papers were relevant to the target of this review. After summarizing the approaches presented in this paper, it can be observed that a majority of blink detection methods based on deep learning primarily rely on CNN, RNN, and transfer learning as their fundamental frameworks. Consequently, this review predominantly focuses on investigating advancements made in these three domains.
How many the amount of reviewed paper? does the total of its enaough for concluding results? How many research work for each CNN, RNN, transfer learning?
2. To assist readers in understanding the significance of the findings, consider including additional contextual information.
3. (line 80)-. Compared with non-learning methods, machine learning has improved its robustness and generalization ability to some extent. However, it is difficult for machine learning methods to find an optimal feature extraction method and feature dimension to extract features as input to the classifier. Improve this sentence to make it easier to understand. for example, add examples or cases, even better if there are references that support it.

Cite this review as
Anonymous Reviewer (2025) Peer Review #2 of "A review of deep learning in blink detection (v0.1)". PeerJ Computer Science

·

Basic reporting

1- Grammar and Clarity: The article requires grammar checks, as some sentences are unclear and the flow is occasionally missing. Proofreading is needed for lines 34-35, 77-78, and 99, as examples
2- Scientific Language: The author should use more scientific language. Refer to lines 67-68, 100, and 135-136 for examples of where this is needed.
3- Ambiguous Sentences: Sentences in lines 84-85 are ambiguous and should be revised for clarity.
4- Rewriting Needed: Lines 103-109 should be rewritten for better clarity and coherence.
5- Abbreviations should be capitalized upon first use. For example, "Support Vector Machine (SVM)" should be used.
6- Lines 86-88: Write out the full term before its abbreviation. For instance, "Convolutional Neural Networks (CNN)" should be used instead of "CNN(convolutional neural networks)".
7-Lines 84-98 should be organized into a coherent paragraph.
9-Paragraph Title: The title for the paragraph starting at line 118 is ambiguous. Consider changing it to “Motivation and Scope of the Review.”
10-Blink Prediction (line 111) vs. Detection: The distinction between eye blink prediction and detection is clear but should be highlighted more effectively. Blink detection involves identifying blinks in real-time or recorded data, while prediction involves forecasting future blinks based on historical data and physiological signals

Experimental design

1-Applications of Blink Detection: The survey should include applications of blink detection in various fields such as medicine, transportation, security, and for people with disabilities. Consider adding a table summarizing fields, relevant studies, deep learning techniques used, and benchmarking results.

2-Deep Learning Models: The focus on three widely used deep learning models needs clarification. The criteria for choosing these models and their prevalence in published studies should be provided. Mention other complex models such as Dual Embedding Video Vision Transformer (DE-ViViT), and other variations of CNN such as ResNet, and AlexNet.

3-Survey Methodology: The section on survey methodology (lines 139) should be enhanced with a table summarizing the number of papers published each year for each approach reviewed, compared to the total number of papers on eye blink detection and other approaches like machine learning or augmented devices.
4-Challenges and Limitations: The paper should discuss challenges faced by deep learning approaches in blink detection, including issues with training, testing, validation, hyper-parameter tuning, and the need for high-performance computing. The cost of time and memory used by deep learning should be highlighted against the cost of using augmented devices for eye blink detection.

Validity of the findings

1--Contact vs. Non-Contact Blink Detection: The distinction between contact and non-contact blink detection methods needs clarification. Contact methods should be defined as those relying on devices directly attached to the eyes (e.g., special lenses or glasses with infrared sensors) or indirectly through techniques like electrooculography (EOG), which detects electrical potentials.
2- Application of Convolutional Neural Network in Blink Detection, this section is missing some research on CNN related networks such as, AlexNet, ResNet, also there are no reported studies during 2024 and there are a less number of recent studies.
3- Paragraph started at line 275, discuss how the researches construct a fusion model of CNN and RNN to detect eye blinks so I think the two approaches that need to be one section named with CNN and RNN Neural Networks started with ones that use CNN only, the second with ones use RNN only and the third with ones use both.
4- Application of transfer learning in blink detection section only has definitions there is no mentioning studies that are using this approach in eye-blink detection and summarized table of theses studies along with their reported accuracies.
5-The section titled “Common datasets” should have a table with links to data sets and data set size and its availability as open/closed access and how many papers used this dataset to consider its effectiveness in the filed and any notes on it.
6-The authors mentioned that The evaluation indicators for eye blinks detection are precision, recall, F1 score, what are about another metrics such as accuracy, AUC-ROC, etc..
7-The study does not mention any future challenges/Limitations regarding using deep learning in an eye-blinking detection
8- The study does not have any practical implementations of testing theses approaches regarding each other and reporting their benchmarking results.

Cite this review as

---

## Round 0.2 · Minor Revisions

· Academic Editor

Minor Revisions

Dear authors,

Thank you for submitting your revised manuscript. Feedback from the reviewers is now available. One of the reviewers thinks that revised paper can be accepted in this form. However it is not recommended that your article be published in its current format. We strongly recommend that you address the issues raised by Reviewer 1 and Reviewer 3 and resubmit your paper after making the necessary changes.

Best wishes,

Reviewer 1 ·

Basic reporting

Reviews on work incorporating transformers, unsupervised learning etc can also be included to address a broad class of audience.

Also this will address RQ3: What are the different models applied in deep learning-based blink detection?

Experimental design

In many places the references are cited in different ways. Eg line no 453 onwards a change in style of citing reference is noted. Follow the same format

Validity of the findings

work incorporating transformers, unsupervised learning etc can also be included

Additional comments

nil

Cite this review as
Anonymous Reviewer (2025) Peer Review #1 of "A review of deep learning in blink detection (v0.2)". PeerJ Computer Science

Reviewer 2 ·

Basic reporting

The improvement you are doing is good

Experimental design

Already good

Validity of the findings

Already good

Additional comments

Thank you for your efforts in addressing my previous concerns. The improvements made to the manuscript have effectively answered all my questions. I appreciate the clarity and depth of the revisions.

Cite this review as
Anonymous Reviewer (2025) Peer Review #2 of "A review of deep learning in blink detection (v0.2)". PeerJ Computer Science

·

Basic reporting

no comment

Experimental design

no comment

Validity of the findings

no comment

Additional comments

1-The commonly used datasets should be in a table with available links, total number of examples, etc.The description of Datasets through the text should be in a summary paragraphs not in a ordered list.
2- In Table 1, the literature should be the total number of papers that are using the modality not listed them through the table as presented.

Cite this review as

---

## Round 0.3 · accepted · Accept

· Academic Editor

Accept

Dear Authors,

Thank you for revising the paper. The reviewers thik that your paper can be accepted in this final form. The paper has been sufficiently revised and is now ready for publication.

Best wishes,

Reviewer 1 ·

Basic reporting

no comment

Experimental design

no comment

Validity of the findings

no comment

Additional comments

no comment

Cite this review as
Anonymous Reviewer (2025) Peer Review #1 of "A review of deep learning in blink detection (v0.3)". PeerJ Computer Science